# PanoWorld-X: Generating Explorable Panoramic Worlds via Sphere-Aware Video Diffusion

**Yuyang Yin** [* 1] **Hao-Xiang Guo** [* 2] **Fangfu Liu** [3] **Mengyu Wang** [1] **Hanwen Liang** [4] **Eric Li** [2] **Yikai Wang** [5] **Xiaojie Jin** [1] **Yao Zhao** [1] **Yunchao Wei** [1 6]

## Abstract

Achieving a complete and explorable 360-degree visual world is a cornerstone of immersive content creation. While recent advances in video generation have achieved impressive results, they follow a **2D-aware paradigm** that treats content generation as transitions of 2D pixels, lacking an intrinsic understanding of the physical 3D world, resulting in frequent geometric inconsistencies. To achieve an explorable and physical-consistent visual world, the generation process should shift to a **3D-aware paradigm**: the visual content is governed by the physical relationships of the entire 3D environment together with 3D motion signals. However, under this setting, the conventional modeling methods and control signals, such as spatial attention computation in a 2D space, become unsuitable and ineffective. To address this, we propose **PanoWorld-X** for **explorable immersive scene video generation**. Our framework is built on the panoramic representation, which naturally maps a 3D scene into a standard format and provides an ideal basis for consistency. Specifically, we first develop a data curation pipeline to produce high-quality and large-motion 3D scene evolution with movement trajectories. To achieve precise control, we design the Exploration Panoramic Plücker Embedding (PPE), a guidance signal tailored for 3D motion. Furthermore, leveraging the spherical geometric properties of panoramic data, we propose a sphere-aware attention mechanism, which can capture true geometric adjacency by reprojecting features onto a spherical surface. Extensive exper-

iments demonstrate that PanoWorld-X achieves superior performance in motion range, control precision, and visual quality, underscoring its potential for real-world applications.

## 1. Introduction

The physical world is a 360-degree, fully explorable spatial environment where observers can perceive their surroundings from any angle. As an observer moves, the visual scene dynamically adapts to their changing position. Replicating these characteristics in the digital domain is critical to support diverse applications, such as the creation of immersive virtual reality (VR) environments (Yang et al., 2024a; Xanthidou et al., 2024) for human users and the development of simulated training spaces (Sora) for embodied intelligence (Liu et al., 2024c; Ma et al., 2024) and autonomous driving agents (Mao et al., 2023). Thus, the generation of immersive and explorable virtual worlds has emerged as a prominent research focus.

Towards this challenging objective, two primary research paths have emerged. The first focuses on large-scale video generation models (Yang et al., 2024c; Kong et al., 2024; Wang et al., 2025). Taking Sora (Sora) as a prime example, these models show promise in simulating environments. However, they are fundamentally constrained by a pixel-centric approach, viewing generation as transitions within 2D distributions over time. To compensate for the lack of explicit spatial grounding, these models depend on additional carefully engineered modules (Liu et al., 2024a; Yu et al., 2024) to enforce geometric structure and temporal consistency, which inevitably increases architectural complexity. As illustrated in Fig. 1, methods based on a purely 2D-aware paradigm often suffer from spatial geometry inconsistencies, primarily due to the temporal forgetting inherent in sequential generation.

An alternative strategy involves leveraging the panoramic format, which, by its nature, maps an entire 3D world into a unified 360° × 180° representation. Despite this inherent advantage, current panoramic video generation meth-

---

[*]Equal contribution [1]Beijing Jiaotong University [2]Skywork AI [3]Tsinghua University [4]University of Toronto [5]Beijing Normal University [6]Beijing Academy of Artificial Intelligence. Correspondence to: Yikai Wang <yikaiw@outlook.com>, Yunchao Wei <yunchao.wei@bjtu.edu.cn>.

*Proceedings of the 43$^{rd}$ International Conference on Machine Learning*, Seoul, South Korea. PMLR 306, 2026. Copyright 2026 by the author(s).

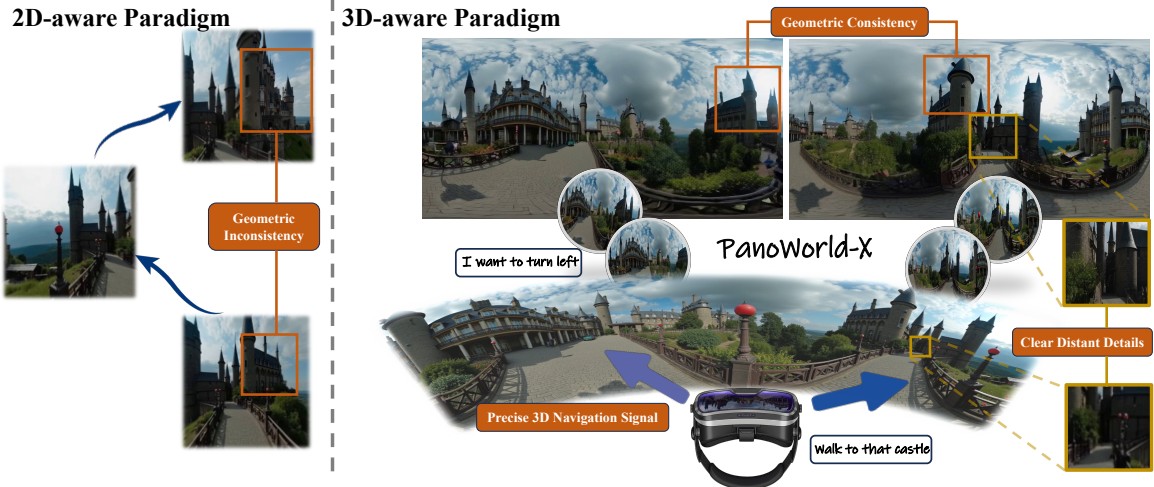

*Figure 1.* Previous methods often view scene generation as a 2D pixel-level transition, which lacks 3D physical consistency. In contrast, PanoWorld-X is specifically designed for panoramic scene video generation by leveraging the projectivity of 3D environments onto panoramas. By enforcing 3D spherical constraints, it ensures superior spatiotemporal consistency during dynamic world exploration. By incorporating precise 3D navigation signals, it enables the synthesis of high-fidelity distant content, facilitating seamless and immersive world exploration.

ods (Wang et al., 2024a; Tan et al., 2024; Li et al., 2024) predominantly treat these images as a special 2D modality. By merely fine-tuning 2D diffusion models, they fail to exploit the 3D spherical priors essential for spatial generation.

Therefore, we argue that the field must shift from a **2D-aware paradigm to a 3D-aware paradigm for immersive scene generation**, governed by two core principles: (1) **3D-Signal Driven State Evolution**: The synthesis of emerging content must be deterministically conditioned on the current state and explicit 3D movement signals. Unlike ambiguous 2D cues, such as text prompts (Xing et al., 2025b) or optical flow (Wang et al., 2024a), precise 3D signals provide the necessary spatial constraints to ensure that the environment evolves in strict accordance with the intended movement trajectory. (2) **3D Projection-aware Visual Fidelity**: Visual updates must strictly reflect actual content transformations within a 3D spherical space. By adhering to 3D projection rules, the generated panoramic sequences maintain rigorous geometric continuity and spatial coherence, effectively preserving the structural integrity of the scene across the entire spherical field of view.

To bridge these gaps, we introduce **PanoWorld-X**, a novel framework for **generating high-quality, explorable panoramic scene videos under movement signals control.** To support this task, we first establish a **systematic data curation pipeline** using Unreal Engine (UE), which enables the synthesis of panoramic content synchronized with complex exploration paths in photorealistic 3D environments.

From the perspective of 3D control, while 6-DoF signals provide the fundamental movement parameters, they lack the granularity to directly guide the displacement of individual pixels. To bridge this gap, we introduce **Exploration Panoramic Plücker Embedding (PPE)**. By adapting previous Plücker coordinates to spherical geometry, PPE serves as an inherently 3D representation that models each pixel as a directional ray in 3D space. This approach effectively transforms global trajectory signals into dense, pixel-level geometric guidance, enabling significantly more precise and physically consistent control over the generated content.

To fully exploit the inherent spherical geometry of panoramic data, we propose **Sphere-Aware Attention**. In standard perspective image or video generation, neighboring regions often contain related information, which leads pretrained models to focus more on nearby tokens. However, panoramic data exhibit distinct characteristics. For example, the left/right edges and polar regions in the equirectangular projection(ERP) panoramas are physically connected in 3D space but spatially separated in 2D projection images. Simply treating panorama data as perspective data when fine-tuning pre-trained models leads the models to underestimate their strong semantic correlations, resulting in reduced coherence. To address this issue, we adhere to the fundamental principle of panorama data by projecting ERP panoramas onto a spherical surface and calculating spatiotemporal spherical distances between patches. Sphere-Aware Attention then acts as a parallel branch that only attends to tokens within a spherical distance threshold, thus enhancing the neglected implicit correlations on the sphere. We find that this approach enhances generation quality and improves global consistency.

We conduct comprehensive experiments to evaluate the pro-

posed framework. The results demonstrate that, compared to existing panoramic video generation models and camera controllable generation models, our PanoWorld-X exhibits superior performance in various aspects, including generation quality, motion range, and control precision. In summary, our key contribution includes:

- We introduce PanoWorld-X for generating explorable and trajectory-controllable panoramic scene videos. Unlike previous 2D-aware methods that focus on pixel changes, our framework treats generation as a 3D-aware process determined by explicit 3D movement signals and spherical geometry constraints.

- We propose a novel panorama dataset preparation pipeline in Unreal Engine and collect over 11K panoramic videos paired with trajectories. To the best of our knowledge, this is the largest synthetic panorama dataset featuring diverse scenes, extensive motion, and comprehensive text and trajectory annotations.

- We introduce Exploration Panoramic Plücker Embedding for effective control in panoramic generation and Sphere-Aware Attention, which leverages the intrinsic spherical connectivity of panoramic data to enhance the spatiotemporal consistency of the generated outputs.

## 2. Related Work

**Video Diffusion Models.** Early research on video diffusion models (Blattmann et al., 2023; Guo et al., 2023; Chen et al., 2023) primarily focused on extending pre-trained image diffusion models (Rombach et al., 2022) to video generation. These studies achieved dimensional extension by incorporating temporal interactions into the UNet architecture of image diffusion models. However, the limited scale of both models and datasets constrained the quality of the generated results. More recently, Diffusion Transformers (DiTs) (Peebles & Xie, 2023) were introduced, with authors demonstrating that Transformer-based (Vaswani et al., 2017; Dosovitskiy et al., 2020) generators exhibit superior scalability compared to UNet-based architectures. Over the past year, several DiT-based text-to-video models (Yang et al., 2024c; Kong et al., 2024; Wang et al., 2025) have emerged. Their successful scaling has significantly improved the performance in various aspects, including generated duration, motion amplitude, and temporal consistency, thus providing a more robust prior for a wide range of downstream tasks(Yin et al., 2023; Liang et al., 2024b; Xing et al., 2025a; Li et al., 2026a).

**View-Controllable Video Generation.** Generating video that matches a sequence of changed views is a critical step toward creating a virtual world and applying it to scenarios such as agent systems. Existing research can be broadly categorized into two main types. The first one (Wang et al., 2024c; He et al., 2024; Liang et al., 2024a; Bahmani et al., 2024; Liu et al., 2024a; Sun et al., 2024; Yu et al., 2024) focuses on incorporating camera extrinsic parameters as additional input signal into pretrained video generation models, allowing the generated video to have perspective changes according to the position and orientation of the camera. The second type (Valevski et al., 2024; Che et al., 2024) aims to generate video games, achieving video-game-like interactivity. The viewpoint changes in these works are controlled by simulating inputs from devices such as keyboards. However, these methods face significant limitations due to the constrained information provided by perspective-view videos. They struggle to generate complete scenes while maintaining 3D consistency and cyclic consistency, leading to notable instability in the generated scenes.

**Panorama Generation Models.** Benefiting from the rapid advancements in 2D image generation, diffusion-based panorama image generation models (Zhang et al., 2024; Li & Bansal, 2023; Wu et al., 2023; Yang et al., 2024b; Feng et al., 2023; Ye et al., 2024; HY-World et al., 2026) has achieved significant results. Despite the 360-degree visible nature of panoramas, physically occluded content (e.g., what lies around the next corner) remains difficult to obtain. Consequently, Panoramic video is necessary to capture broader spatial information, facilitating the construction of a comprehensive world model. Recent works have explored dynamic panoramic video generation under different input settings. 360DVD (Wang et al., 2024a), 4K4DGen (Li et al., 2024), GenEX (Lu et al., 2024) and subsequent works (Liu et al., 2024b; Yang et al., 2025; Dong et al., 2025; Xing et al., 2025b) synthesize panoramic videos from a single text prompt or a panoramic image, while Imagine360 (Tan et al., 2024), 360Anything (Wu et al., 2026), CubeComposer(Li et al., 2026b) and Beyond the frame (Luo et al., 2025) synthesize panoramic videos from perspective-view inputs. However, these approaches are unable to generate content with significant viewpoint progression, restricting the scope of world content generation to relatively limited areas. Furthermore, these methods fail to achieve precise and controllable 360-degree worlds, which hinders the ability to interact effectively with the generated worlds. Therefore, our objective is to generate explorable panoramic videos that incorporate extensive spatial movement.

## 3. Method

To achieve high-quality panoramic scene video scene generation with free exploration, we first curate the PanoExplorer dataset in Section 3.2. We then introduce the Exploration Panoramic Plücker Embedding as a 3D control signal in Section 3.3. Furthermore, to better exploit the underlying 3D geometric properties and enhance panoramic video

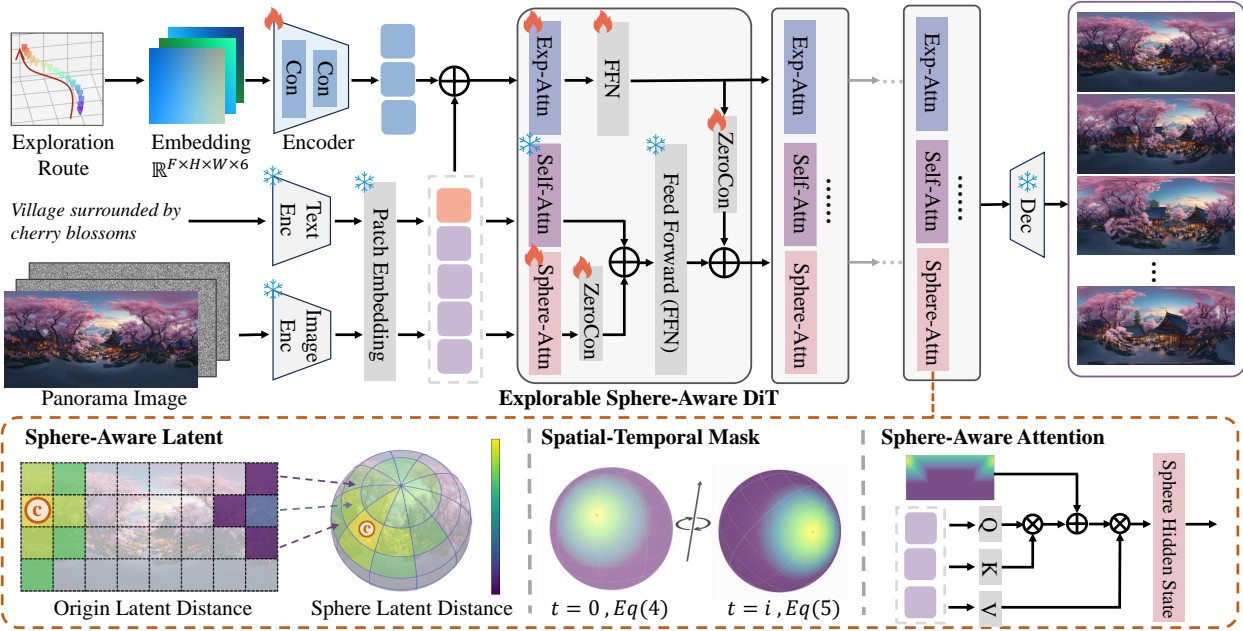

*Figure 2.* **PanoWorld-X Framework.** Given an initial panorama and a 3D trajectory, we first encode the signal into Exploration Panoramic Plücker Embeddings. These embeddings are integrated into the Explorable Sphere-Aware DiT to enable controllable panoramic generation. A key component is the Sphere-Aware Attention, which enforces spatiotemporal coherence by modeling token relationships via spherical geometry. By bridging boundary gaps (e.g., left and right edges) that are distant in 2D but adjacent in 3D, our framework ensures true geometric continuity across the 360-degree scene.

generation quality, we propose a Sphere-Aware Attention mechanism in Section 3.4.

### 3.1. Preliminary

**Video Diffusion Model.** The latest text-to-video generation models adopt latent-space multimodal DiT architectures, which consist of three key components: text encoders (Radford et al., 2021; Raffel et al., 2020) for encoding textual prompts, a 3D-VAE (Yu et al., 2023; Yang et al., 2024c) for video data compressing and tokenization, and a Transformer-based generator. As the core component, the generator is composed of numerous transformer blocks. It takes flattened video tokens and text tokens as inputs, modeling video and textual information through 3D self-attention (Yang et al., 2024c; Kong et al., 2024). However, this architecture presents certain challenges. On one hand, it incurs high computational and memory costs, making it more difficult to introduce additional control signals to guide the generation process. On the other hand, despite its global receptive field, the model sometimes overlooks essential long-range dependencies, resulting in decreased geometric consistency in the generated outputs. For panoramic video, which exhibits substantial distribution differences compared to common visual data, these issues become even more pronounced.

### 3.2. Data Curation

Collecting panoramic videos with real-world exploration trajectories is often hindered by high labor costs and uncontrollable environments. To address this, we introduce the **PanoExplorer Dataset**, which, to our knowledge, is the largest synthetic panoramic dataset available. It features **116,759 high-quality video sequences** across diverse scenes with extensive motion patterns and comprehensive text-trajectory annotations. Built entirely in Unreal Engine, our scalable pipeline ensures high-fidelity simulation and precise trajectory control through four key stages:

*Step 1: Data Collection in Unreal Engine.* Unreal Engine enables flexible data preparation across diverse scenes, perspectives, and content. Using 504 high-fidelity 3D scenes, we cover varied indoor and outdoor environments with different weather and lighting, forming the basis for trajectory-based video recording.

*Step 2: Exploration Route Sampling.* To generate plausible and coherent camera paths, we first extract walkable surfaces and apply Delaunay triangulation to construct a non-overlapping triangular mesh. Path candidates are generated by selecting random points, computing the shortest path via Dijkstra's algorithm, and applying Laplacian smoothing to eliminate abrupt turns. We retain only trajectories exceeding 18 meters, where each route is defined as

$Er^T = (x^T, y^T, z^T, \alpha^T, \beta^T, \gamma^T)$. Here, $(x, y, z)$ denote spatial coordinates and $(\alpha, \beta, \gamma)$ represent the Euler angles (yaw, pitch, and roll).

*Step 3: Collision Detection.* To prevent geometry clipping and object intersections, we implement a bounding box proxy algorithm. By simplifying complex scene geometry into 3D bounding boxes, we efficiently simulate each trajectory step-by-step and discard any intersecting paths, balancing spatial accuracy with computational speed.

*Step 4: Data Annotation and Quality Filtering.* We ensure dataset quality through two filtering stages: (1) Automatic Filtering: We employ Video-LLaMA3 (Zhang et al., 2025) to assess videos based on detailed quality, semantic information, and motion richness, and subsequently filter out low-quality content. (2) Manual Assessment: The first frame of each video is manually screened to eliminate samples with poor rendering quality or missing details. Finally, Video-LLaMA3 automatically annotates videos to support text-controlled and multimodal tasks. For a more detailed statistical analysis of the dataset, please refer to Appendix A.

### 3.3. Exploration Panoramic Plücker Embedding (PPE)

In the pursuit of high-precision control, purely 2D conditioning signals, such as text prompts or optical flow, are inherently ambiguous and insufficient to impose strict geometric constraints on scene generation. While traditional 3D motion signals, such as 6-DoF parameter (comprising $XYZ$ translations and Euler angles: pitch, yaw, roll) serve as the fundamental movement parameters from datasets. However, these signals are essentially sparse global descriptors that lack the geometric granularity required to directly guide the displacement of individual pixels. Previous methods for view-controllable video generation (He et al., 2024; Liang et al., 2024a) typically rely on explicit camera intrinsic and extrinsic parameters to construct 3D Plücker rays as an auxiliary geometric proxy.

However, our task faces the fundamental challenge that panoramic datasets generally lack these geometric parameters. Even if one assumes equivalent intrinsic/extrinsic parameters for a panoramic frame, directly applying standard Plücker embeddings—designed for perspective pinhole models—introduces significant geometric inconsistency. In a pinhole model, direction vectors are derived via a linear transformation of the pixel grid $d_{u,v} = K^{-1}[u, v, 1]^T$. This assumption fails in Equirectangular Projection (ERP), which maps a 360° spherical field onto a rectangular grid. For example, a horizontal pixel shift corresponds to a fixed angular change in a pinhole model; in contrast, the same pixel shift in ERP represents a large angular span near the equator but negligible movement near the poles. Standard Plücker rays therefore treat these stretched polar regions as distinct spatial locations, forcing the model to process many inconsistent geometric instructions for a single physical point at the North Pole. This leads to excessive computational pressure and fragmented textures. To address this, we propose Exploration Panoramic Plücker Embedding (PPE), which reformulates the ray representation through two core components:

**1) Spherical Manifold Mapping:** We redefine the ray representation within a spherical manifold to ensure topological coherence. For a pixel $(u, v)$ in an ERP image of size $W \times H$, we map it to longitude $\lambda = (\frac{2u}{W} - 1)\pi$ and latitude $\phi = (\frac{2v}{H} - 1)\frac{\pi}{2}$. The unit direction vector $d_{u,v,t}$ is computed as:

$$d_{u,v,t} = R_t \cdot \left[\cos\phi\sin\lambda \ \ \sin\phi \ \ \cos\phi\cos\lambda\right] \qquad (1)$$

where $R_t$ is the rotation matrix from the 6DoF signal $Er_t$. This ensures polar pixels naturally converge to the same vector $[0, \pm1, 0]^T$, allowing the model to aggregate redundant features and maintain geometric stability at the poles.

**2) Metric-Anchor Normalization:** To handle diverse 3D assets with varying absolute scales, we introduce a normalization strategy to unify the movement magnitude. Using the 75th percentile of the scene's metric depth $D_{75}$ as a reference, we calculate the normalized translation $\hat{T}_t$:

$$\hat{T}t = \frac{T_t}{D75} \cdot \sigma_{anchor} \qquad (2)$$

where $\sigma_{anchor} = 10$ cm. The final 6-channel PPE is formulated as a unified 3D geometric representation, where each pixel is encoded by its corresponding ray direction and moment in 3D space: $\text{PPE}_{u,v,t} = (d_{u,v,t}, \hat{T}_t \times d_{u,v,t})$. This representation embeds pixel-wise information directly into a physically grounded 3D space, enabling dense and geometry-aware control.

To inject these features, we follow the ControlNet (Zhang et al., 2023) philosophy by replicating the original DiT blocks to create an Exploration-Aware Attention (Exp-Attn) module. The PPE features are compressed via 3D convolutions and fused with the original spatiotemporal tokens through zero-linear layers, preserving high-fidelity synthesis while achieving precise navigation control.

### 3.4. Sphere-Aware Attention

Previous studies (Wang et al., 2024a; Lu et al., 2024) fine-tune perspective generative models (Blattmann et al., 2023; Rombach et al., 2022) on panoramas, treating it as a process of adapting models to a specialized data domain. However, this straightforward fine-tuning strategy neglects the inherent 3D properties of panorama data, as the pixels in panoramas are actually distributed in a sphere, which has significantly different geometric characteristics compared to plane geometry. Therefore, we introduce the following spatiotemporal distance representation based on spherical

geometry, which has a crucial impact on the subsequent attention process.

In existing generative model frameworks, images or videos are encoded into a series of tokens. Previous 3D self-attention mechanisms enable the capture of spatially proximate tokens to gather more contextual information. Typically, different positional distances are determined based on planar geometry, where the distance between two points $p_1 = (x_1, y_1)$ and $p_2 = (x_2, y_2)$ is measured using the Euclidean distance:

$$d_{\text{Euclidean}}(p_1, p_2) = \sqrt{(x_2 - x_1)^2 + (y_2 - y_1)^2}. \quad (3)$$

While in panoramic ERP images, pixels distributed on a spherical surface introduce significantly different properties. For example, the point on the leftmost in Fig. 2 is physically connected with the points on the rightmost and those near the polar region in the spherical coordinates. However, when unfolded into a flattened image, these areas exhibit large spatial coordinate distances as in Eq. (3). Therefore, we introduce the spherical distance (Great-circle Distance) to accurately measure the distance between two points on the sphere. For an ERP image of width $W$ and height $H$, a pixel at coordinates $(x, y)$ can be converted to spherical coordinates $(\theta, \phi)$. Building on geometric priors, we reproject the ERP panorama image onto a spherical surface and recalculate the distances. For two points with spherical coordinates $(\theta_1, \phi_1)$ and $(\theta_2, \phi_2)$, the spherical distance $d_{\text{spherical}}$ is given by the Haversine formula:

$$d_{\text{spherical}}(p_1, p_2) = 2R \cdot \arcsin\left(\sqrt{\sin^2\left(\frac{\phi_2 - \phi_1}{2}\right)} \right.$$
$$\left. + \cos(\phi_1)\cos(\phi_2)\sin^2\left(\frac{\theta_2 - \theta_1}{2}\right)\right)$$
$$(4)$$

From a temporal perspective, movement in the real world can be interpreted as the rotation of the panoramic sphere in various directions. We define a rotation matrix $\mathbf{R}^{t_i}(\alpha, \beta, \gamma)$ that rotates a point on the sphere by Euler angles $\alpha$ (yaw), $\beta$ (pitch), and $\gamma$ (roll) at time $t_i$ relative to its initial orientation at time $t_0$. For a point $\mathbf{p}^{t_i} = (\theta, \phi)$ on the sphere, the rotated point $\hat{\mathbf{p}}^{t_i}$ relative to $t_0$ is given by

$$\hat{\mathbf{p}}^{t_i} = \mathbf{R}^{t_i}(\alpha, \beta, \gamma) \cdot \mathbf{p} = \mathbf{R}_z^{t_i}(\alpha) \cdot \mathbf{R}_y^{t_i}(\beta) \cdot \mathbf{R}_x^{t_i}(\gamma)\mathbf{p}, \quad (5)$$

where $\mathbf{R}_z(\alpha)$, $\mathbf{R}_y(\beta)$, and $\mathbf{R}_x(\gamma)$ are the rotation matrices around the z-axis, y-axis, and x-axis, respectively. So we measure the point $\mathbf{p}_1^{t_0}$ and $\mathbf{p}_2^{t_i}$ distance by first rotate to $\hat{\mathbf{p}}_2^{t_i}$ then calculate the sphere distance given in (4).

Based on the above analysis, we introduce **Sphere-Aware Attention** (Sphere-Attn) to enhance the generative model's

perception of spherical geometric features. Specifically, we redefine the distances between latent representations based on their positions on the spherical surface and attempt to enhance the response of closely located latent blocks. We set the attention mask to 1 for pairs of spatial-temporal positions whose spherical distance is below a predefined threshold $\tau$, indicating regions that require mutual reinforcement; otherwise, the mask value is set to 0. In this way, the spatial-temporal attention mask selectively activates interactions between nearby regions on the spherical surface.

Following standard 3D self-attention, video tokens are projected into query $Q$, key $K$, and value $V$. The Sphere-Aware attention mask $M$ is then incorporated as an additive bias to the scaled dot-product attention, such that only spatial-temporal token pairs permitted by the mask contribute to the attention weights. The resulting attention weights are normalized via softmax and used to aggregate the value features.

As illustrated in Fig. 2, we employ a parallel mechanism. The video path simultaneously feeds into both the original self-attention block and the newly trained Sphere-Aware attention block. To ensure the Sphere-Aware attention block does not affect the initial training outcomes, a zero-initialized linear block is utilized. Compared to a sequential design, this parallel structure avoids significantly disrupting the output of each layer, which would otherwise lead to reduced training efficiency and increased difficulty in convergence.

## 4. Experimentation

### 4.1. Implementation Detail

Since our model is based on a video diffusion framework, we select the advanced diffusion transformer model, CogVideoX-5B-I2V (Yang et al., 2024c). We fine-tuned the model to generate 49 frames with a resolution of $480 \times 720$. Due to the panoramic nature, the width-to-height ratio must adhere to 1:2. Therefore, we resized the output frames to $480 \times 960$. Compared to training a model with a native 1:2 aspect ratio, this post-processing resizing approach maximally preserves the prior knowledge of the original model. The model is trained on 8 A100 GPUs about 10,000 iterations. During the inference stage, we first generate a panorama image using a text prompt with FLUX (Labs, 2024), leveraging the panorama LoRA released by Yang (Yang et al., 2024b). Subsequently, we input the first image along with specific action signals into our model.

### 4.2. Evaluation Datasets and Metrics

To evaluate the performance of explorable panoramic scene video generation, we randomly select 200 panoramic videos

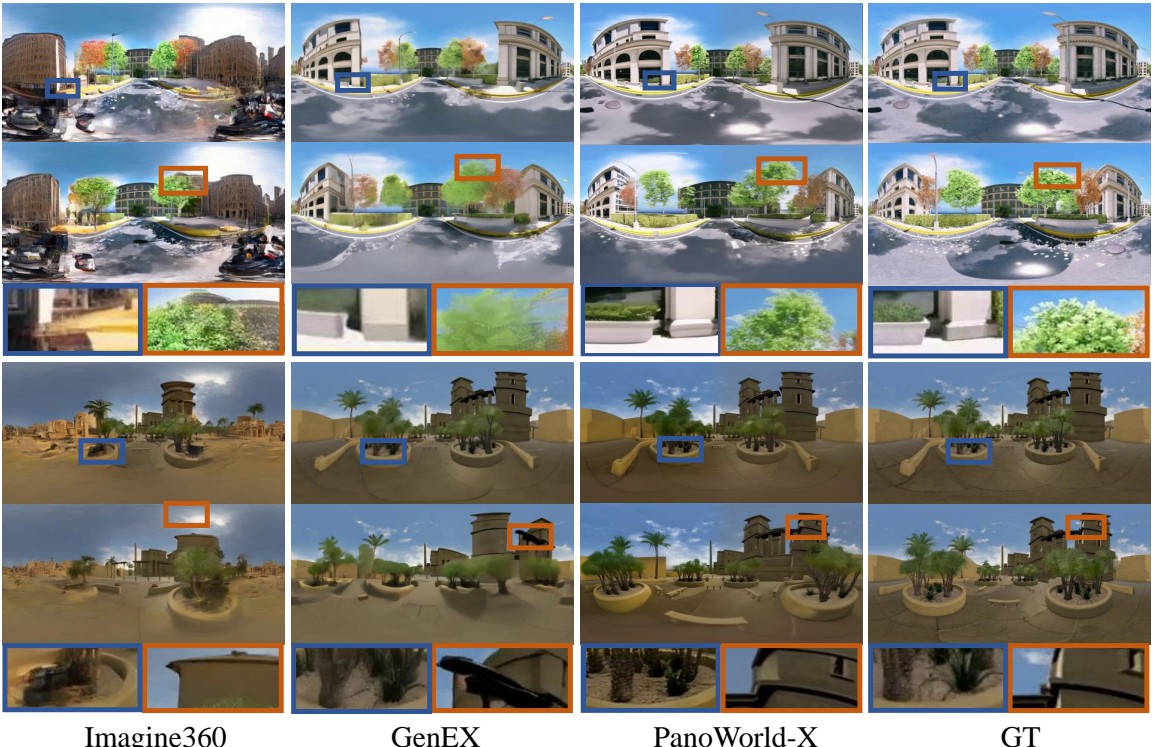

|  |  |  |  |
| :---: | :---: | :---: | :---: |
| Imagine360 | GenEX | PanoWorld-X | GT |

*Figure 3.* Qualitative comparisons with panoramic video generation models on generated keyframes. Our results exhibit superior detail clarity and geometric coherence throughout the movement. **Additional video visualization results are provided in the supplementary materials.**

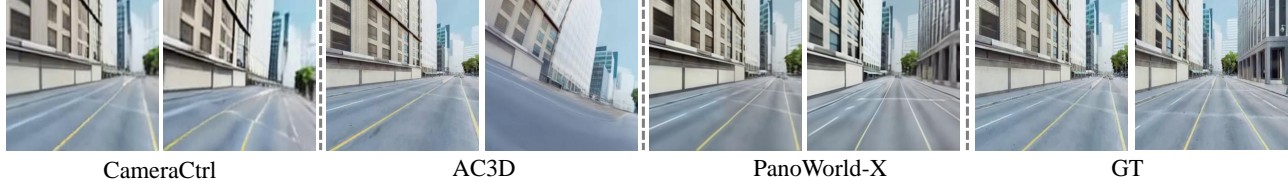

|  |  |  |  |
| :---: | :---: | :---: | :---: |
| CameraCtrl | AC3D | PanoWorld-X | GT |

*Figure 4.* Qualitative comparisons with camera controllable methods on cropped perspective views from panoramas. Our results demonstrate more precise camera control and generation quality.

from our curated dataset. Each panoramic video includes a movement trajectory. We compare the generated videos with ground-truth video clips using multiple metrics:(1) Pixel-level visual quality is measured using PSNR, SSIM, and LPIPS. (2) Visual quality and temporal coherence are assessed using Frechet Inception Distance (FID) (Heusel et al., 2017) and Frechet Video Distance (FVD) (Unterthiner et al., 2019). (3) Exploration route control precision is evaluated using Rotation Error ($R_{err}$) and Translation Error ($T_{err}$) metrics, as introduced by (He et al., 2024). These metrics compute the camera extrinsic parameters in comparison to the ground truth camera pose. To accommodate varying output lengths, the mean of these metrics is calculated rather than their sum.

### 4.3. Quantitative Results

**Comparison with Panoramic Video Generation Models.** We evaluate the performance of our proposed methods by comparing them with state-of-the-art Panoramic Video generation models. Specifically, we select three baseline methods for comparison:(1) 360DVD (Wang et al., 2024a) uses a trainable 360-Adapter to extend standard T2V (Guo et al., 2023) models to the panorama domain. (2) Imagine360 (Tan et al., 2024) is a model designed to convert perspective videos into panoramic videos. We crop perspective videos from the ground truth videos and evaluate their generation capabilities. (3) GenEX (Lu et al., 2024) is a stable video diffusion (Blattmann et al., 2023) based model for Panoramic Video generation. Based on its official check-

*Table 1.* Comparison of Panoramic Video Generation and Camera Controllable Generation Models.

| Models | PSNR ↑ | SSIM ↑ | LPIPS ↓ | FID ↓ | FVD ↓ | $R_{err}$ ↓ | $T_{err}$ ↓ |
|---|---|---|---|---|---|---|---|
| *Panoramic Video Generation Models* | | | | | | | |
| 360DVD (Wang et al., 2024a) | 10.66 | 0.35 | 0.73 | 111.85 | 2049.21 | – | – |
| Imagine360 (Tan et al., 2024) | 11.62 | 0.39 | 0.591 | 66.73 | 1830.46 | – | – |
| GenEX (Lu et al., 2024) | 16.12 | 0.59 | 0.42 | 42.22 | 1113.72 | – | – |
| PanoWorld-X | **19.34** | **0.63** | **0.24** | **28.01** | **467.18** | – | – |
| *Camera Controllable Generation Models* | | | | | | | |
| CameraCtrl (He et al., 2024) | 11.56 | 0.38 | 0.61 | 108.12 | 2017.95 | 0.097 | 0.245 |
| AC3D (Bahmani et al., 2024) | 13.77 | 0.49 | 0.52 | 41.98 | 842.29 | 0.081 | 0.087 |
| PanoWorld-X Perspetive | **16.76** | **0.56** | **0.42** | **38.63** | **586.51** | **0.061** | **0.073** |

*Table 2.* Ablation Study on Individual Components.

| Models | PSNR ↑ | SSIM ↑ | LPIPS ↓ | FID ↓ | FVD ↓ | $R_{err}$ ↓ | $T_{err}$ ↓ |
|---|---|---|---|---|---|---|---|
| w/o Data Normalization | 17.11 | 0.55 | 0.32 | 40.37 | 751.18 | 0.114 | 0.102 |
| w/o PPE | 16.30 | 0.53 | 0.36 | 38.71 | 769.42 | 0.102 | 0.152 |
| w/o Sphere-Aware Attention | 17.59 | 0.56 | 0.27 | 29.96 | 492.98 | 0.069 | 0.076 |
| Full model | **19.34** | **0.63** | **0.24** | **28.01** | **467.18** | **0.061** | **0.073** |

*Table 3.* Comparison of PSNR on different viewing directions with orther method.

| Method | Front | Left | Right | Back |
|---|---|---|---|---|
| 360DVD | 8.94 | 9.37 | 9.11 | 9.25 |
| Imagine360 | 16.06 | 10.97 | 10.95 | 9.64 |
| GenEX | 15.89 | 15.77 | 15.99 | 16.17 |
| Ours | **16.76** | **16.67** | **17.04** | **17.31** |

point, it only supports image-to-video generation and lacks a controllable video generation checkpoint. Therefore, we only compare its video quality with our proposed methods.

The qualitative comparison shown in Fig. 3 demonstrates the high-quality generation capability of our model. Imagine360 struggles to consistently generate stable content on both sides, often leading to detail degradation. GenEX tends to produce blurry images during the generation process. In contrast, our method not only maintains excellent detail in the generated images but also achieves a significantly wider range of camera movement compared to other approaches. Additionally, Tab. 1 confirms that our method outperforms previous methods across all evaluation metrics. To further evaluate the quality of perspective videos compared to panoramic video generation methods, we divide the panoramic video into four main sections: front, left, right, and back. Specifically, we convert the panoramic image into four perspective images. As shown in Tab. 3, our method outperforms previous state-of-the-art approaches in terms of generation quality across all directions. This

demonstrates that our results not only achieve better global generation quality but also surpass previous methods in fine local details.

**Comparison with Camera Controllable Generation Models.** To evaluate our explorable route-controllable Panoramic Video generation, there are no existing works with the same setting for direct comparison. As a result, we compare our approach with camera-controllable generation models. Since these models are not trained on panoramic datasets, we crop our results into perspective view videos and input the ground truth perspective view images into camera-controllable models to ensure a fair comparison. We select two state-of-the-art methods for this evaluation: CameraCtrl (He et al., 2024) and AC3D (Bahmani et al., 2024). We evaluate the generation results from two perspectives. First, in terms of image quality, as shown in Tab. 1, our method surpasses previous approaches. The quality of details is also evident in the comparison provided in Fig. 4. Second, regarding controllability, CameraCtrl often struggles to achieve precise control over content, typically allowing only very limited movement. While AC3D shows improvement in controllability compared to earlier methods, it still performs poorly when handling complex trajectories. In contrast, our method demonstrates a significant enhancement in controllability compared to previous approaches.

For more comparison results, please refer to Appendix B.

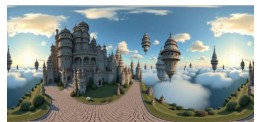 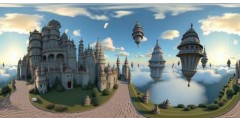 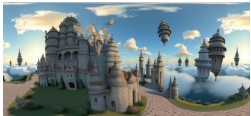 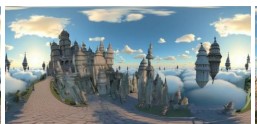 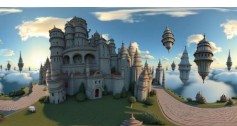

| Input (first frame) | W/o Normalization | W/o PPE | W/o Sphere Attention | Full Model |

*Figure 5.* Ablation study. The data normalization strategy improves the range of motion. The PPE enhances control precision. The sphere-aware attention significantly enhances the quality of panoramic details, particularly in terms of left-right consistency.

*Table 4.* User study on in-the-wild panoramic videos. We report the percentage of times each method is ranked first by participants.

| Model | Graphics Quality ↑ | Frame Consistency ↑ | End Continuity ↑ | Content Distribution ↑ |
|---|---|---|---|---|
| 360DVD | 10.5% | 11.2% | 10.1% | 8.8% |
| Imagine360 | 11.3% | 12.6% | 10.9% | 9.4% |
| GenEX | 26.4% | 27.2% | 29.5% | 23.2% |
| Ours | **51.8%** | **47.0%** | **56.5%** | **62.6%** |

**User Study on In-the-wild Data.** Since in-the-wild panoramic videos do not provide ground-truth references, we further conduct a user study to evaluate perceptual quality. Following the protocol of 360DVD, we invite 30 participants to compare the generated panoramic videos from different methods. For each test case, participants are asked to rank all methods from four aspects: graphics quality, frame consistency, end continuity, and content distribution. We report the percentage of times that each method is ranked first in Table 4. Our method achieves the highest preference across all four dimensions. These results indicate that our approach produces more visually appealing, temporally stable, and spatially coherent panoramic videos under real-world scenarios where reference-based metrics are unavailable.

### 4.4. Ablation Study.

We investigate the importance of the proposed components from three key aspects:**(1) Metric-Anchor Normalization:** By normalizing the dataset stride scale, we establish a consistent spatial metric across diverse environments. This enhances the model's sensitivity to 3D navigation signals and facilitates more significant camera movements. This anchoring ensures stable velocity perception, preventing visual drift in expansive scenes while maintaining structural integrity in constrained spaces. **(2) Exploration Panoramic Plücker Embedding (PPE)**: Controlling panoramic generation with sparse signals (e.g., raw 6-DoF) is weak due to a lack of spatial granularity. Furthermore, traditional Plücker Embeddings do not adapt to panoramic topology, leading to poor control at poles and boundaries, which causes imprecise trajectories and artifacts. In contrast, our PPE is tailored for spherical geometry, enabling precise forward movement and arbitrary rotation. Quantitatively, it significantly reduces $R_{err}$ and $T_{err}$, ensuring the output strictly adheres to input signals. **(3) Sphere-Aware Attention**: This

mechanism enhances generation by reprojecting features onto the spherical surface, enabling the model to capture true 3D geometric relationships rather than relying on flat 2D approximations. As a result, it achieves consistently higher quantitative scores across all metrics in Tab. 2, while also producing finer details and more realistic textures in Fig. 5.

## 5. Limitations.

Our current model does not natively support high-resolution panoramic video generation, mainly due to the resolution constraints of the underlying video diffusion backbone. In practice, higher-resolution results can be obtained by applying post-processing video super-resolution methods(Wang et al., 2021; Xie et al., 2025). For long panoramic video generation, we adopt a multi-segment generation strategy to extend the temporal duration. However, when the target video becomes very long, temporal degradation and accumulated inconsistencies may still occur. We note that resolution, video length, and inference speed are common challenges in the broader video generation field, and are largely orthogonal to the panoramic modeling problem studied in this work. These aspects are expected to improve progressively as stronger video generation backbones and more efficient inference techniques become available.

## 6. Conclusion

We introduce PanoWorld-X, a novel framework that shifts panoramic video generation from a 2D pixel-transition paradigm to a 3D-consistent synthesis paradigm. To support this task, we first curate a large-scale, high-fidelity dataset featuring diverse scenes and synchronized 3D trajectories. Building upon this, we address the inherent limitations of 2D modeling including uncontrollable movement and geometric inconsistency by introducing the Exploration Panoramic Plücker Embedding (PPE) for precise trajectory control and a sphere-aware attention mechanism to enforce global spatiotemporal coherence. Our evaluations demonstrate that PanoWorld-X significantly outperforms existing state-of-the-art methods in both control fidelity and visual quality, which underscores its potential as a scalable solution for immersive 360-degree content creation.

# Acknowledgements

This work was supported by the Fundamental Research Funds for the Central Universities (No. 2025YJS048), the National Natural Science Foundation of China (No. 92470203), the BAAI Scholar Foundation (No. Z251100008125015), the National Natural Science Foundation of China (No. U23A20314), the Beijing Natural Science Foundation (No. L242022), the General Program of the National Natural Science Foundation of China (No. 62576043), and the Fundamental Research Funds for the Central Universities (No. 2243100024).

# Impact Statement

This paper presents work whose goal is to advance the field of Machine Learning. There are many potential societal consequences of our work, none which we feel must be specifically highlighted here.

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

# Supplementary Materials for
## *PanoWorld-X: Generating Explorable Panoramic Worlds via Sphere-Aware Video Diffusion*

## A. Details of the Curated Dataset

### A.1. Visualization of Data Pairs

This subsection provides an overview of the curated dataset. Our PanoExplorer Dataset comprises 116,759 high-quality data pairs. Each pair consists of a panoramic video sequence and its corresponding exploration route. We present several samples from the dataset to provide an intuitive understanding of its structure and stylistic characteristics in Fig. A.

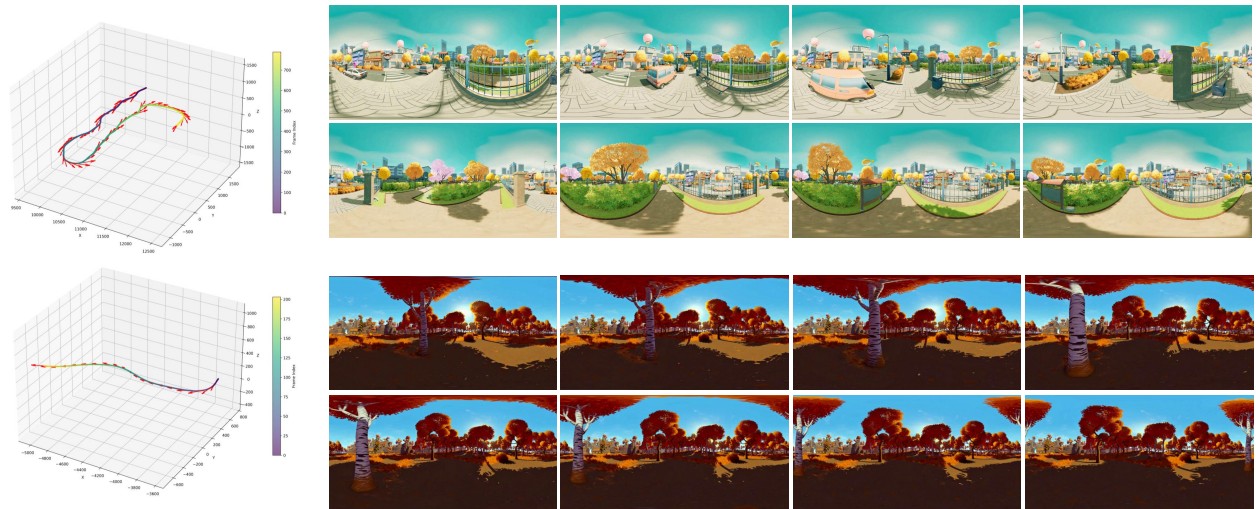

*Figure A.* **PanoExplorer Datasets.** We present two samples from the datasets. The sequences, arranged from left to right and top to bottom, represent video frames evolving along the timeline. The left side illustrates the visualized exploration routes.

### A.2. Process of Generating Text Captions with Examples

Our caption is generated by Video-LLaMA3 (Zhang et al., 2025), which is the state-of-the-art VLM model for video caption. We give Video-LLaMA3 the prompt like 'Generate a detailed, temporally-aware caption for this panoramic video. Describe the environment (indoor/outdoor, natural/urban, time of day, weather, lighting......), key subjects (objects/people/animals with appearances, actions, and interactions......), and camera dynamics (static/smooth rotation/fast panning, direction, perspective, and horizon tilt......). Note temporal changes (new subjects entering, lighting shifts......) and spatial details (foreground/background layers......).'

An example of caption: "The video showcases a vibrant, alien landscape with tall, mushroom-like trees and glowing plants. The scene is bathed in a deep blue light, creating an otherworldly atmosphere. The camera slowly pans to the right, revealing more of the fantastical environment. The sky is filled with stars, and a large, glowing moon hangs in the distance. The ground is covered in lush, blue grass, and there are large, rocky formations scattered throughout the landscape. The overall effect is one of wonder and exploration." This example demonstrates that our captions provide comprehensive descriptions (approximately 100 words) while effectively conveying multiple aspects of the scene.

### A.3. Comparison with Other Panorama Datasets

We compare our *PanoExplorer* dataset with existing panoramic datasets. Web360 (Wang et al., 2024a) is one of the early panoramic generation datasets, sourced from 360-degree camera videos on YouTube. Imagine360 (Tan et al., 2024) and PanoWan (Xia et al., 2025) expanded on this by increasing the volume and diversity of the data. However, these datasets primarily focus on outdoor natural landscapes, which inherently limit the range of motion in the scenes and lack trajectory information. 360-1M (Wallingford et al., 2024) is currently one of the larger panoramic datasets, also filtered from 360-degree videos on YouTube. However, it lacks textual annotations and trajectory data, requiring reliance on additional

models like Colmap or Dust3R (Wang et al., 2024b) for reconstruction. This introduces potential inaccuracies, making it unsuitable for tasks requiring precise trajectory control. DiffPano (Ye et al., 2024) utilizes the Habitat Simulator to render a panoramic video dataset based on the Habitat Matterport 3D (HM3D) dataset (Ramakrishnan et al., 2021). Similar to our approach, DiffPano employs synthetic data generation. However, it is limited to indoor scenes, with motion restricted to movement within and between rooms, and the dataset size remains relatively small.

Compared to previous datasets in Tab. A, our *PanoExplorer* dataset offers several advantages:

1. **Richer Scene Diversity** – Our method enables the collection of diverse panoramic videos by leveraging various 3D assets.

2. **More Controllable Trajectories** – Real-world captured videos lack trajectory information, requiring estimation through additional models, which can introduce errors. In contrast, virtual environments allow precise trajectory control and easy storage of motion data, ensuring accurate paired samples.

3. **Larger-Scale Dataset Collection** – Unlike previous approaches that rely on costly real-world data collection and manual annotation, our method generates a significantly larger dataset at a fraction of the cost.

*Table A*. Comparison of panoramic datasets.

| Dataset | Samples | Indoor | Outdoor | Diversity | Motion | Trajectory | Text | Origin |
|---------|---------|--------|---------|-----------|--------|------------|------|--------|
| Web360 | 2,114 | × | ✓ | Low | Low | × | ✓ | Real-World |
| Imagine360 | 10,744 | × | ✓ | Medium | Medium | × | ✓ | Real-World |
| PanoWan | 13,000 | ✓ | ✓ | Medium | Medium | × | ✓ | Real-World |
| 360-1M | 1,076,592 | ✓ | ✓ | High | High | × | × | Real-World |
| DiffPano | 8,508 | ✓ | × | Low | Low | × | ✓ | Synthetic |
| Ours | **116,759** | ✓ | ✓ | **High** | **High** | ✓ | ✓ | Synthetic |

# B. More Visualization of Panoramic Video Generation

## B.1. Evaluation of zero-shot generalization on in-the-wild images

To verify our model's zero-shot generalization capability, we use diverse in-the-wild images to generate panoramic videos. The first images are either pre-existing or generated. As illustrated in Fig. B, we evaluate our model using images from diverse environments, including outdoor, indoor, real-world scenes, and synthetic scenarios. The generated results demonstrate high-quality details and accurate geometric representation. The results demonstrate the ability to handle various exploration routes, such as moving forward, turning left, and turning right.

## B.2. Comparison on the Test Set

In this section, we compare our method with state-of-the-art panoramic video generation methods on test sets to evaluate the quality of the generated outputs. As illustrated in Fig. C, our results exhibit clearer details and better image quality.

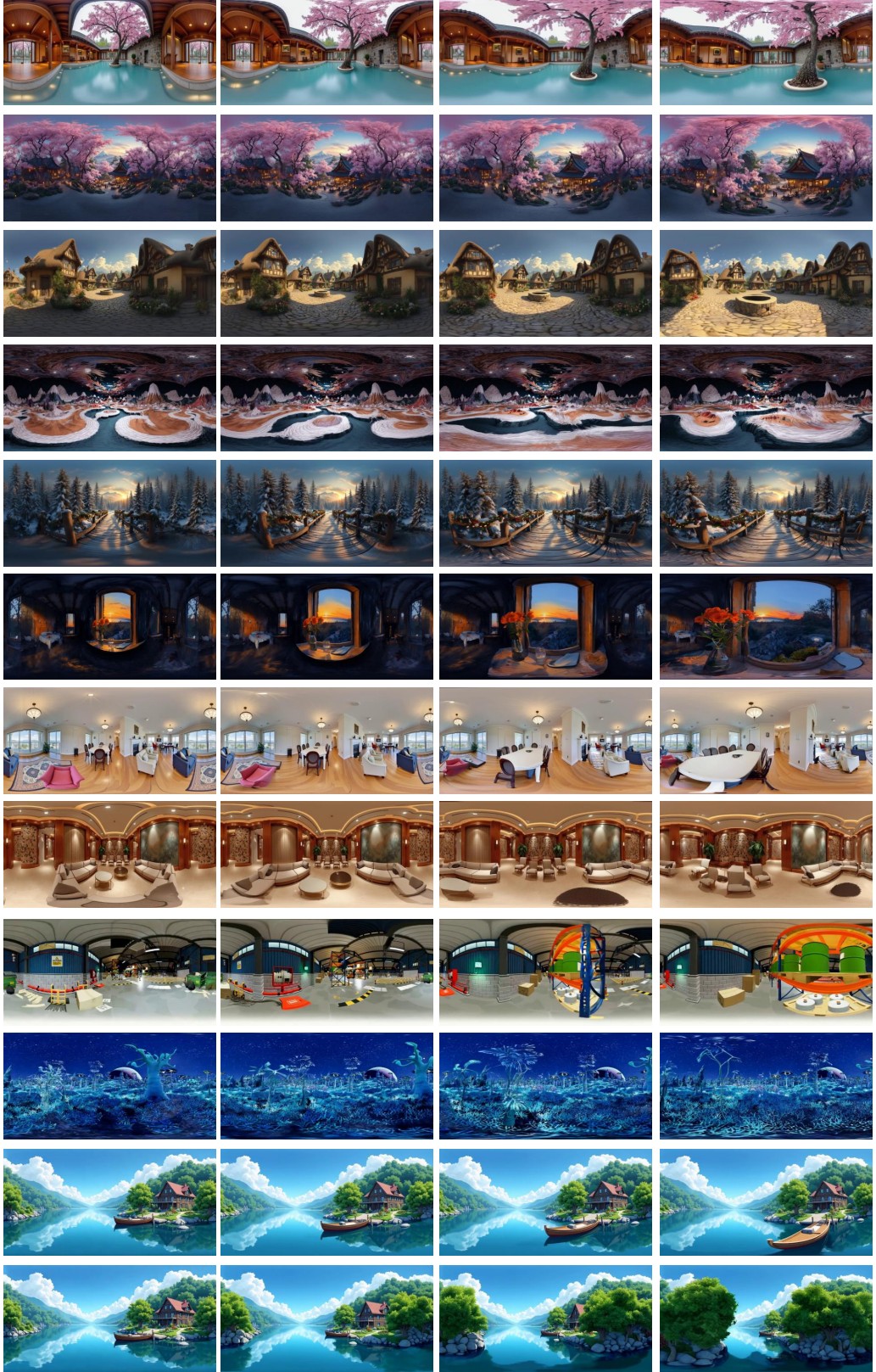

*Figure B.* Visualization of zero-shot generalization on in-the-wild images.

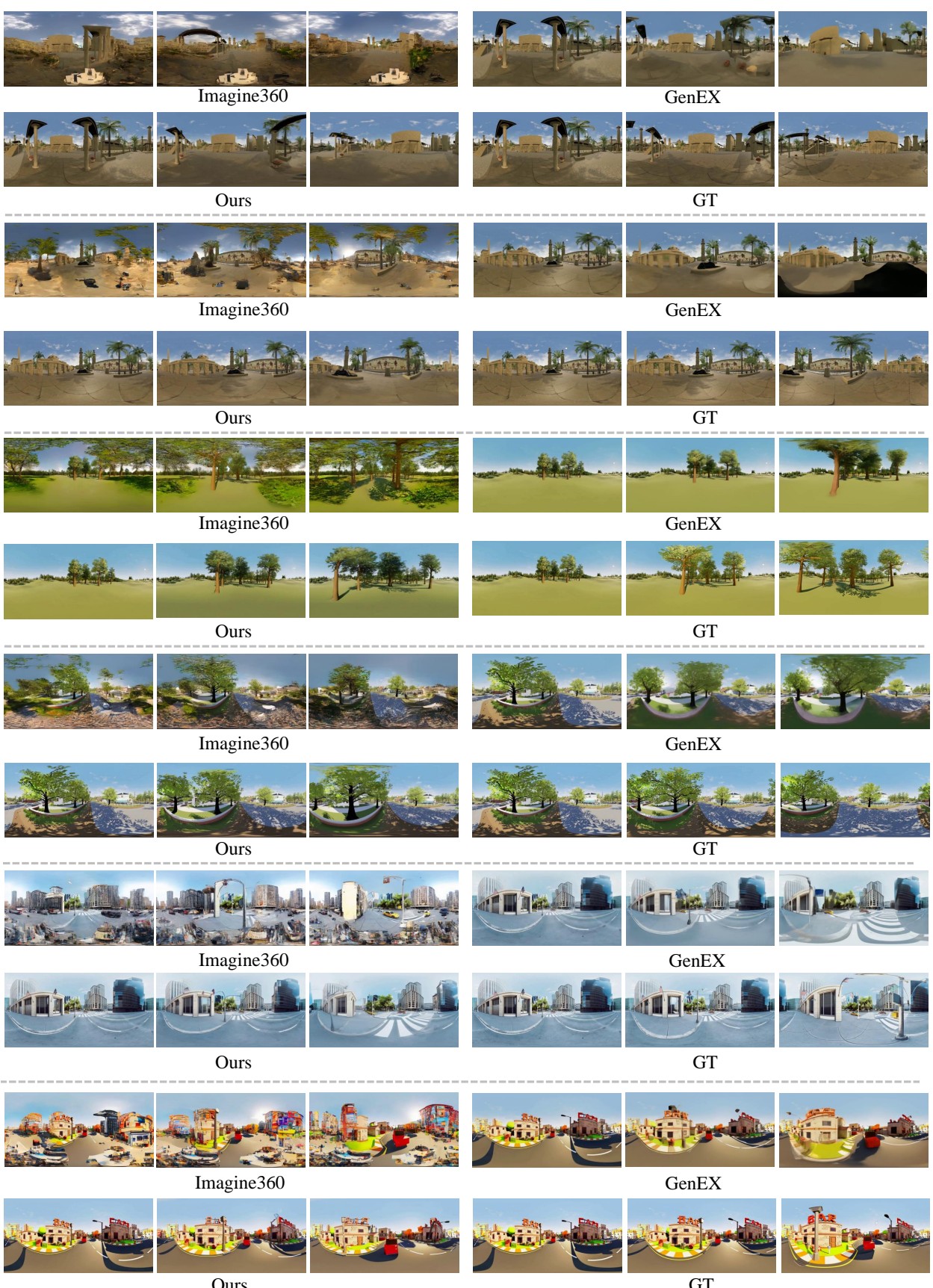

*Figure C.* Visualization of the comparison with panoramic generation methods on the test sets.

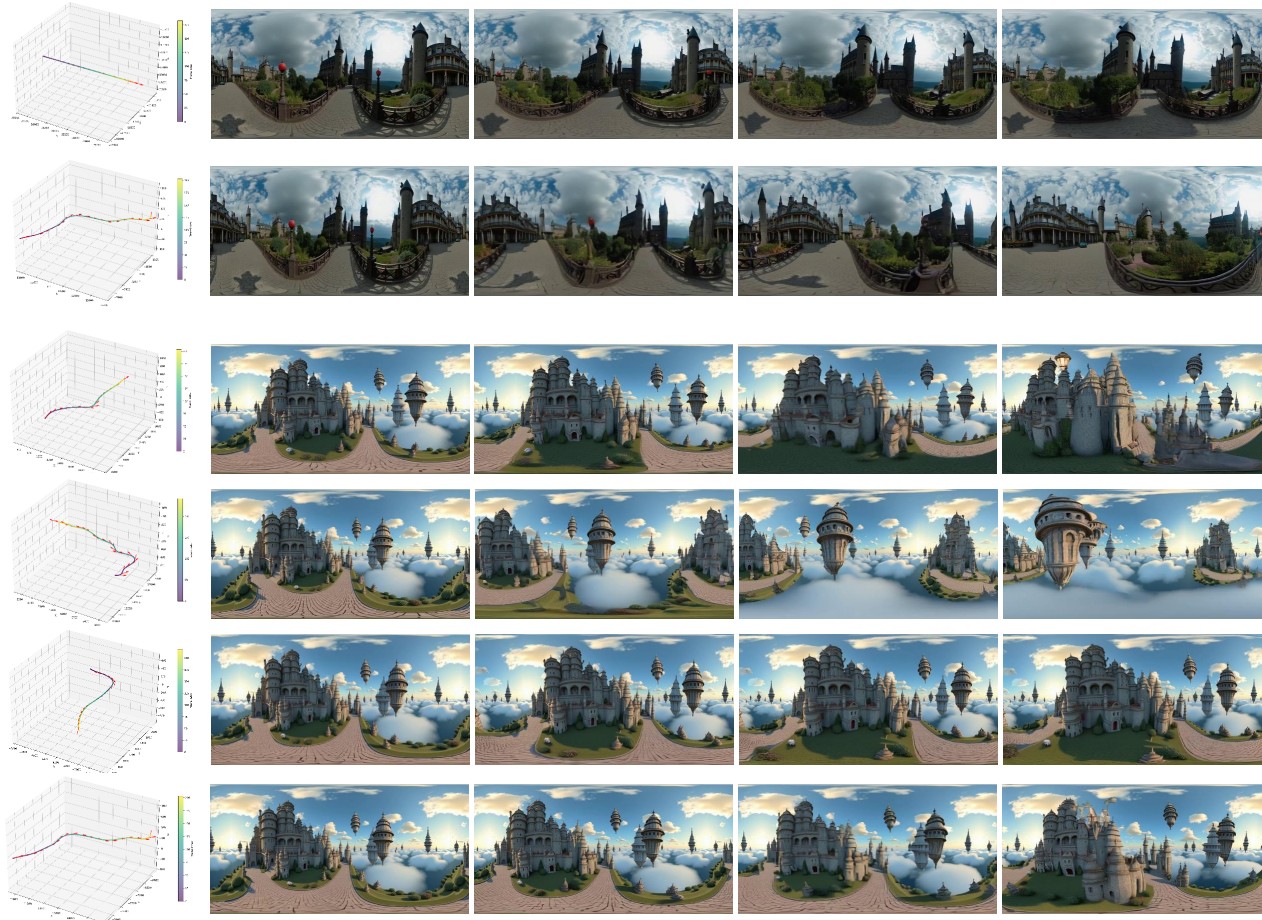

*Figure D.* Visualization of various exploration routes on a single image.

### B.3. Visualization of diversity exploration routes

In this section, as shown in Fig. D, we use various exploration routes on a single image to demonstrate our model's capability to handle diversity trajectory and its precise camera control functionality.

## C. Comparison with Camera-Controllable Generation Models

In this section, we compare our method with state-of-the-art camera-controllable generation methods on test sets to assess the quality of trajectory control. Since previous methods were not trained on panoramic datasets, we crop perspective videos from panoramic videos for fair comparisons.

As shown in Fig. E, our results align more closely with the ground truth compared to previous methods and demonstrate a greater range of camera motion.

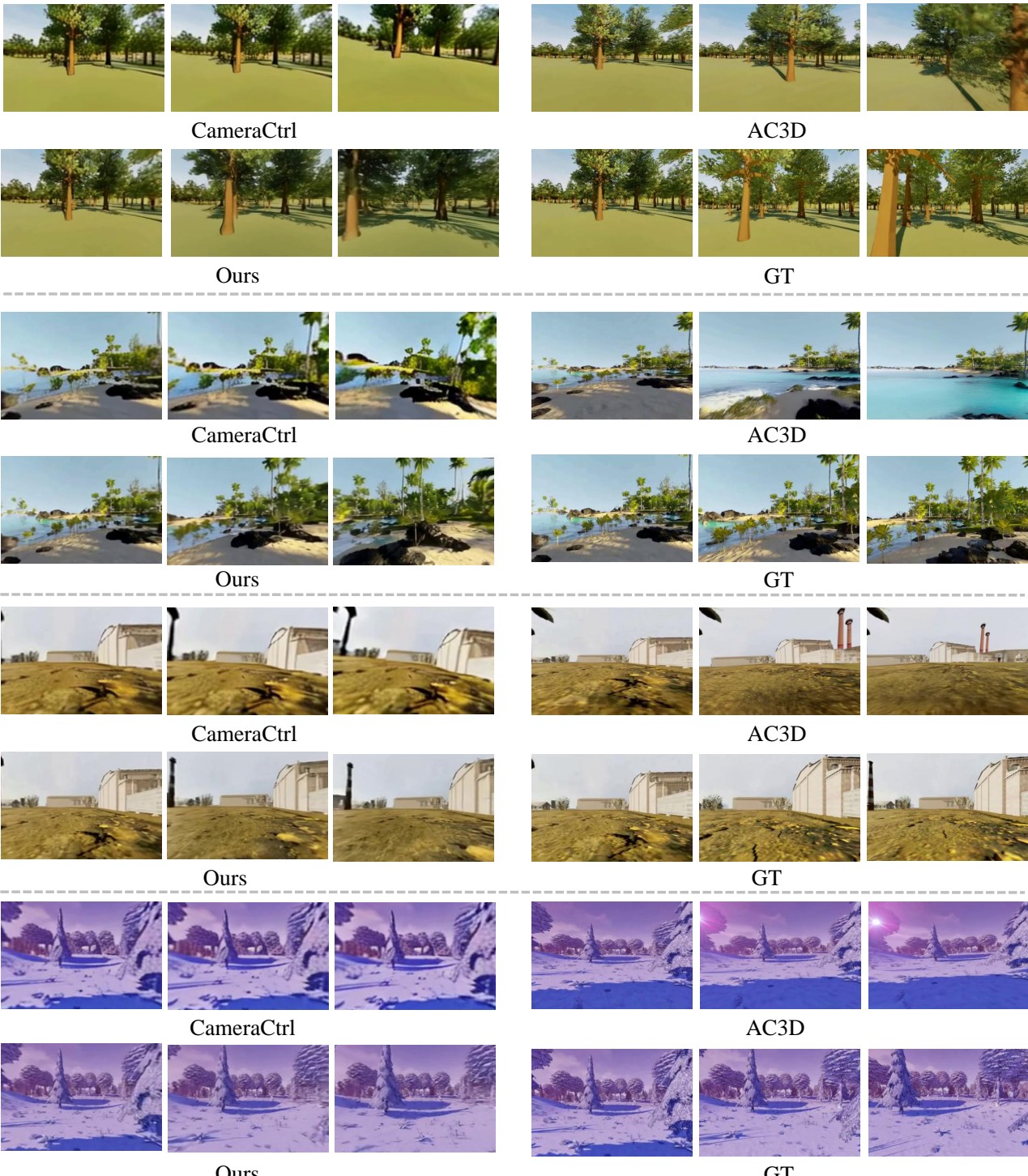

*Figure E.* Visualization of the comparison with camera-controllable generation methods on the test sets.

