# OpenReview forum: "PanoWorld-X: Generating Explorable Panoramic Worlds via Sphere-Aware Video Diffusion"
_ICML.cc/2026/Conference — ICML 2026 spotlight_

### Official Review · Reviewer_hAHM · 2026-02-17

**Soundness:** 3
**Presentation:** 3
**Significance:** 2
**Originality:** 3
**Overall Recommendation:** 5
**Confidence:** 5

**Summary:**

This paper proposes PanoWorld-X, a trajectory-controlled panorama video generation framework. On the modeling side, the authors propose sphere-aware attention and Panoramic Plucker Embedding; on the dataset side, this work introduces a large-scale synthetic 360 video dataset rendered in Unreal Engine. Experiments demonstrate improved performance over existing methods. However, issues remain in the scope of baselines and the generalizability.

**Compliance With Llm Reviewing Policy:**

Affirmed.

**Final Justification:**

Please see the rebuttal acknowledgement.

**Key Questions For Authors:**

1. Could the authors provide any controlled experiments or failure analyses specifically targeting dynamic scenes (e.g., moving people or vehicles)?

2. The paper reports some in-the-wild results, however, qualitative results are missing. Could the author explain why don't collect a real-world evaluation dataset (e.g., using 360 videos from YouTube of capture by 360 cameras, then run calibration methods such as ViPE) for evaluation? Such experiments would be beneficial.

3. Since the dataset native resolution is higher (2K×4K), is the 480×960 output strictly limited by the base model, or are there other bottlenecks that prevent scaling to higher resolutions? If GPU memory consumption is the major bottleneck, is it possible to distill the model into a few-step video generation model (e.g., by self-forcing) and perform streaming generation?

4. Regarding the data synthesis pipeline, how are the 3D environments collected? Do the scenes contain mountain or uneven terrains where walking simulation is more difficult? Can the synthesis framework generalize to non-static scenes?

**Limitations:**

yes

**Strengths And Weaknesses:**

## Strengths

The paper tackles an important problem of trajectory-controlled image-to-360° video generation, an important topic in immersive content creation and AR/VR.

The proposed sphere-aware designs are well motivated and justified by the experiments.

The dataset itself is a contribution, but its usage might be limited due to the lack of dynamic objects.

## Weaknesses

**1. Limited ability to handle dynamic content**. A central limitation of the work is its focus on static scenes. Given that the model is only trained on static videos, the model cannot generate dynamic elements such as human walking. The authors could potentially use real-world dataset proposed in [1,2] or synthetic video with dynamic contents following in pipeline proposed in [3] with Unreal Engine (but for render 360 videos). Since real-world immersive environments frequently contain dynamic objects, the current approach may not generalize well beyond static exploratory scenarios.

**2. Missing comparison with recent video-to-360 methods.**   The paper does not compare/discuss against recent video-to-360 panorama generation works such as [1,4], which are highly relevant to this problem setting. In particular, work [4] discusses that [1] outperforms Imagine360, yet Imagine360 is used as a primary reference baseline in this paper. Note that comparison with [4] is not necessary since it is a closed-sourced concurrent work, but discussion would be beneficial.

**3. Limited output resolution and short video length.**  The model generates relatively short clips at 480×960 resolution. While the authors argue that long-video generation and super-resolution are out of scope, these constraints limit the practical usability of the system for immersive viewing.

- [1] Beyond the Frame: Generating 360° Panoramic Videos from Perspective Videos. ICCV, 2025.
- [2] PanoWan: Lifting Diffusion Video Generation Models to 360° with Latitude/Longitude-aware Mechanisms. NeurIPS, 2025.
- [3] ReCamMaster: Camera-Controlled Generative Rendering from A Single Video. ICCV, 2025.
- [4] 360Anything: Geometry-Free Lifting of Images and Videos to 360°. arXiv, 2026.

---

> ### Author Rebuttal · Authors · 2026-03-31
>
> **Visualization: https://panoworld-x.github.io/Video/**
>
> **Q1: Dynamic Content and Dataset**
>
> (1) Challenges with real-world datasets [1,2]. We investigated incorporating [1,2], but these datasets suffer from photographer's hands visible in frames, unstable hand-held camera motion, and camera parameter errors from post-hoc estimation. Our joint training experiments with synthetic + real data showed that such noise made training significantly harder to stabilize and converge.
>
> (2) Dynamic content via [3]. Synthesizing 360° videos with dynamic objects in UE is a promising direction. However, [3] only released the dataset without data preparation scripts, making it difficult to reproduce. We consider this a viable future direction with the community. We have further tested the generalization boundary of our model on dynamic scenes (results on project page).
>
> **Q2: Missing Related Works**
>
> [1, 4] focus on video-to-panoramic-video projection, whereas our task is single-image-driven generation with controllable trajectories. We will add a discussion in the revised paper.
>
> **Q3: Resolution and Video Length**
>
> *(1) Resolution bottleneck.* Scaling resolution is not simply a matter of more compute—it requires architectural advances. The primary bottleneck is that varying resolution breaks the RoPE positional priors learned during pre-training, causing severe quality degradation. Additionally, the noise schedule / SNR calibration and 3D VAE receptive field are both tied to the pre-trained resolution.
>
> *(2) Few-step distillation does not address resolution.* Few-step distillation (e.g., consistency distillation, self-forcing) accelerates sampling by reducing denoising steps, but does not increase output resolution. Distilling a high-resolution model would first require a fully trained high-resolution teacher, which does not yet exist for panoramic video generation at 2K×4K.
>
> *(3) Video super-resolution as a practical, orthogonal solution.* Rather than training at native high resolution, we apply video super-resolution as post-processing—a strategy orthogonal to our generation framework. It leverages powerful existing SR models while fully preserving our generation quality and 3D consistency. Super-resolved results on our project page demonstrate high-quality panoramic videos suitable for immersive HMD viewing.
>
> For longer videos, we adopt an autoregressive extrapolation strategy: consecutive clips share overlapping frames at boundaries, where tail frames initialize the next clip's noise latents, anchoring appearance and motion continuity. This produces continuous sequences of **15–20 seconds** with minimal error accumulation. Results are on our project page.
>
>
> **Q4. About Dynamic**
>
> We add dynamic examples on the project page using motion-related prompts, showing that training on synthetic static data can still generalize to dynamic scenes. Failure cases mainly arise in two situations: (1) deformable objects such as pedestrians, where current image generators often fail to produce a plausible ERP-consistent first frame, limiting subsequent video generation; and (2) large motions, where preserving both realistic motion and ERP geometry is difficult, causing local artifacts and temporal inconsistencies.
>
> **Q5: Real-World Evaluation**
>
> In the revision, we add two complementary real-world evaluations.
>
> First, for visual quality and zero-shot generalization, we build a 100-image benchmark from WEB360 and self-collected panoramas:
>
> | Model | FID↓ | FVD↓ | Q-Align↑ |
> |---|---:|---:|---:|
> | 360DVD | 92.31 | 167.39 | 0.487 |
> | Imagine360 | 87.24 | 135.03 | 0.492 |
> | GenEX | 82.31 | 142.35 | 0.413 |
> | **Ours** | **64.82** | **103.45** | **0.551** |
>
> These results show strong generalization to real-world panoramas.
>
> Second, for real-world trajectory control, we curate 50 videos with reliable trajectories in youtube and provide motion visualizations in the supplementary video.
>
> | Model | Rerr↓ | Terr↓ |
> |---|---:|---:|
> | CamCtrl | 0.134 | 0.384 |
> | AC3D | 0.134 | 0.238 |
> | **Ours** | **0.082** | **0.103** |
>
> We did not previously build a large trajectory-paired real-world benchmark because panoramas do not follow the pinhole model, making calibration methods such as ViPE unreliable after projection conversion. Real 360° videos also contain compression artifacts, stitching seams, and hand-held motion, which introduce strong confounding factors.
>
> **Q6: Data Synthesis Pipeline**
>
> (1) **3D environments:** 504 high-fidelity Unreal Marketplace scenes across 10 categories (Table A), including mountains, forests.
>
> (2) **Uneven terrains:** we use lightweight 3D box collision detection instead of mesh-level parsing, and reduce vertical jitter by elevating the camera above the ground plane with Laplacian trajectory smoothing.
>
> (3) **Dynamic scenes:** Route Sampling and Collision Detection depend only on scene geometry and are agnostic to dynamics. When assets contain animations, the pipeline captures them directly without modification.

---

> > ### Author Rebuttal · Reviewer_hAHM · 2026-04-03
> >
> > Thanks for the authors rebuttal and I believe most of my concerns are addressed. However, given that Argus consistently performs better than Imagine360 as shown in 360Anything CubeComposer (a more recent related work on panoramic video generation), comparing with it could bring benefit to the paper.
> >
> > Besides, I have a follow-up question on the data pipeline. By what portion of the 3D environment assets are paid since paid assets of usually of higher quality? It would be great if the author can share the list of used environment assets.
> >
> > Overall, I decided to increase my rating to accept.

---

> > > ### Author Response · Authors · 2026-04-03
> > >
> > > **Thank you for your feedback and for increasing the rating to `Accept`."**
> > >
> > > We will cite and discuss Argus, 360Anything , CubeComposer and other related works in the camera-ready version. Regarding the data, high-quality paid assets comprise 60%-70% of our environments, further refined by automated and manual screening. We will address the potential disclosure of the asset list upon acceptance, as it requires navigating third-party commercial licenses and completing internal legal and compliance reviews.

---

### Official Review · Reviewer_y8r8 · 2026-03-10

**Soundness:** 2
**Presentation:** 3
**Significance:** 2
**Originality:** 2
**Overall Recommendation:** 4
**Confidence:** 5

**Summary:**

PanoWorld-X is an innovative framework for generating explorable 3D scene videos that shifts the traditional video generation paradigm from 2D pixel transitions to a physically consistent 3D environment. By leveraging a panoramic representation to map 3D scenes into a standard format, the system addresses the common issue of geometric inconsistency found in 2D-based models. To achieve this, the authors developed a specialized data curation pipeline for high-quality, large-motion 3D evolution and introduced the Panoramic Plücker Embedding (PPE) to provide precise guidance for 3D motion trajectories. Furthermore, the framework incorporates a sphere-aware attention mechanism that reprojects features onto a spherical surface to capture true geometric adjacency, overcoming the distortions inherent in flat spatial attention. Experimental results demonstrate that PanoWorld-X significantly outperforms existing methods in motion range, control precision, and visual quality, offering a robust solution for immersive 360-degree content creation.

**Compliance With Llm Reviewing Policy:**

Affirmed.

**Key Questions For Authors:**

see weakness

**Limitations:**

see weakness

**Strengths And Weaknesses:**

Strengths

1. Paradigm Shift to 3D Consistency: Unlike traditional 2D video generation that suffers from geometric warping, PanoWorld-X adopts a 3D paradigm. By utilizing a panoramic representation, it ensures a physically consistent 360-degree environment that maintains structural integrity during camera exploration
2.  Innovative Sphere-aware Attention: The framework effectively addresses the "polar distortion" inherent in equirectangular projections. By reprojecting features onto a spherical surface to capture true geometric adjacency, it eliminates the artifacts typically found at the top and bottom of panoramic maps.
3. High-Precision Motion Control: The introduction of Exploration Panoramic Plücker Embedding (PPE) provides a sophisticated guidance signal. This allows for far more accurate control over complex 3D camera trajectories (rotation, translation, and scaling) compared to standard 2D coordinate systems.
4. Large-Scale Explorability: The specialized data curation pipeline enables the model to handle high-quality, large-motion scene evolutions. This makes it highly suitable for practical applications like VR, digital twins, and immersive virtual tours.

Weaknesses:
1. Resolution and Sampling Constraints: Mapping a full 360-degree scene into a standard 2D format often leads to uneven pixel density. This can result in localized blurring or loss of detail when the camera "zooms" into specific areas of the panorama.

2. Computational Complexity of Spherical Projection: The sphere-aware attention mechanism involves coordinate transformations and feature re-projections. This likely increases the computational overhead and VRAM requirements compared to standard spatial attention, potentially limiting its scalability for ultra-high-resolution output.

3. Handling of Independent Dynamic Objects: While the framework excels at camera-driven scene exploration, its ability to model multiple, independently moving foreground objects (e.g., pedestrians or vehicles) while maintaining global 3D consistency remains unproven.

4. High Dependency on Curated Trajectory Data: The performance is tightly coupled with the quality of the training data pipeline. The model's generalization capabilities may be limited in domains where high-quality 3D scene data with precise movement trajectories are unavailable.

---

> ### Author Rebuttal · Authors · 2026-03-31
>
> **Visualization: https://panoworld-x.github.io/Video/**
>
> **Q1: Video Resolution**
>
> We acknowledge that the current output resolution is a practical limitation. To address this limitation in practice, we have applied  video super-resolution modes  as a post-processing step to upscale the generated panoramic videos. The super-resolved results demonstrate significantly improved visual quality and are showcased on our anonymous project page. Importantly, this enhancement is orthogonal to our core contributions (PPE, Sphere-Aware Attention) and can be seamlessly applied to any panoramic video generation pipeline, making it a general and practical solution for resolution scaling.
>
> **Q2: Computational Complexity of Sphere-Aware Attention**
>
> Indeed, our sphere-aware attention introduces additional memory and computation — a trade-off between performance and cost. Quantitatively, inference memory increases from 5 GB to 8 GB and time from ~200s to ~280s on a single H20 GPU for a 49-frame video. We believe this is worthwhile: it improves not only visual quality and quantitative metrics, but critically enhances left-right boundary continuity on the panoramic sphere (see Supplementary Video4-ablation-sphere-attn.mp4), which is essential for HMD viewing where vertical seams severely degrade immersion.
>
> Regarding ultra-high-resolution scalability, the bottleneck is not our additional branch. Even without it, current base models (CogVideoX, Wan 2.1) cannot train at resolutions like 2K×4K — doing so would break the RoPE positional priors learned during pre-training. As a practical alternative, we apply video super-resolution as post-processing, with results demonstrated on our project page.
>
> **Q4. About Dynamic**
>
> We add dynamic examples on the project page using motion-related prompts, showing that training on synthetic static data can still generalize to dynamic scenes.  Failure cases mainly arise in two situations: (1) deformable objects such as pedestrians, where current image generators often fail to produce a plausible ERP-consistent first frame, limiting subsequent video generation; and (2) large motions, where preserving both realistic motion and ERP geometry is difficult, causing local artifacts and temporal inconsistencies.
>
>
> **Q4: Dependency on Curated Trajectory Data**
>
> (1) Strong zero-shot generalization. As shown in Fig. B and Video1, the same trajectory types (e.g., forward, turn-left) produce geometrically consistent results across vastly different domains — outdoor landscapes, indoor rooms, artistic scenes, and real-world photographs — none of which appear in training.
>
> (2) **Cross-domain trajectory control evaluation.** We design a **relative trajectory accuracy** metric. For each of 20 trajectories, we apply it to 5 in-the-wild images from different domains, estimate poses from the generated videos, and compute `R_err^rel` and `T_err^rel` relative to the input trajectory. The resulting `R_err^rel` is **0.014** and `T_err^rel` is **0.034**, demonstrating that the same trajectory yields nearly identical motion patterns across domains. This confirms that our model does not merely fit the training data distribution, but learns a **domain-agnostic geometric prior** for trajectory-to-motion mapping. We attribute this to our geometry-grounded design (PPE + Sphere-Aware Attention) and the visual priors of CogVideoX-5B preserved by LoRA.

---

> > ### Author Rebuttal · Reviewer_y8r8 · 2026-04-03
> >
> > The authors’ responses address part of my concerns, and I will maintain my score.

---

> > > ### Author Response · Authors · 2026-04-03
> > >
> > > We sincerely thank the reviewer for the positive rating and support. Your recognition is highly motivating for our team, and we will carefully refine the paper according to your feedback for the final submission.

---

### Official Review · Reviewer_dweu · 2026-03-11

**Soundness:** 4
**Presentation:** 4
**Significance:** 3
**Originality:** 3
**Overall Recommendation:** 5
**Confidence:** 4

**Summary:**

The paper proposes PanoWorld-X for generating explorable panoramic scene videos with trajectory control. It makes three contributions: (1) PanoExplorer, a large-scale synthetic dataset of 116K panoramic videos with trajectory annotations, built in Unreal Engine; (2) Exploration Panoramic Plücker Embedding (PPE), which adapts Plücker coordinates to spherical geometry for dense pixel-level 3D control; and (3) Sphere-Aware Attention, which replaces Euclidean distance with great-circle distance in attention masks so that tokens respect true spherical adjacency. The model is built on CogVideoX-5B and evaluated on a held-out test set of 200 videos, comparing against panoramic video generation methods (360DVD, Imagine360, GenEX) and camera-controllable generation methods (CameraCtrl, AC3D).

**Compliance With Llm Reviewing Policy:**

Affirmed.

**Final Justification:**

The rebuttal directly addressed all three of my questions with real numbers. The user study on mixed test-set and in-the-wild inputs, the real-world FID/FVD evaluation, and the latitude-stratified PSNR analysis, which show the largest gains at the poles where the method claims to help most, are all convincing. The authors also acknowledged the 2D vs 3D framing concern and committed to revision. I am raising my score from weak accept to accept.

**Key Questions For Authors:**

Questions for Authors

1. User study. Can you provide a perceptual evaluation, ideally on in-the-wild inputs? This would be the most direct way to validate quality beyond synthetic ground truth. If validated, this would be the highest impact change.

2. Temporal consistency over long sequences. The spherical representation helps with spatial consistency per frame, but how does the model handle longer sequences or multi-hop exploration where temporal drift becomes the dominant issue?

3. PPE at the poles in practice. The convergence of polar pixels to a single direction vector is theoretically clean, but how does generation quality actually look at the poles? Any systematic quality analysis by latitude?

**Limitations:**

Yes

**Strengths And Weaknesses:**

What works:

- The PPE is the strongest contribution. The argument for why standard Plücker coordinates break on ERP is concrete and well-explained: a horizontal pixel shift corresponds to a fixed angular change in a pinhole model but a varying angular span in ERP, and polar pixels get mapped to many disparate geometric instructions for what is physically a single point. The spherical manifold mapping (Eq. 1) handles this cleanly by ensuring polar pixels converge to the same direction vector. This is a specific, transferable insight that anyone working with Plücker embeddings on panoramic data can use.

- The Sphere-Aware Attention is a simple idea but well-motivated and well-ablated. Using great-circle distance instead of Euclidean distance for the attention mask means left/right ERP edges (far in 2D, adjacent on the sphere) can attend to each other. The convergence speed results (Table D) are interesting: sphere-aware attention at 7000 steps matches the baseline at 10,000 steps, suggesting the spherical prior provides genuine inductive bias rather than just a marginal improvement. The threshold ablation (Table E) and dynamic temporal mask ablation (Table F) indicate that the authors have carefully considered the design space.

- The dataset is substantial: 116K videos across 504 UE scenes with trajectory annotations, scene diversity analysis, and trajectory complexity statistics. If released, this could be the most impactful contribution for the community. The data pipeline (walkable surface extraction, Delaunay triangulation, Dijkstra path planning, Laplacian smoothing, collision detection) is well-engineered and reproducible.
- The paper is clean overall. Three contributions, each individually motivated and ablated.

- The qualitative evaluations are well-presented. Figure 5 in particular makes a clear visual case for each component; the left-right boundary consistency comparison (part b) is more convincing than the numbers alone.

Core Concerns:

- Evaluation is entirely on their own synthetic data, with no user study. The 200 test videos are held out from the same UE-rendered distribution, and the zero-shot in-the-wild results (Fig. B) are qualitative only. I recognize that trajectory-paired panoramic video benchmarks are scarce, so direct external comparisons aren't straightforward. But some form of real-world validation is needed: FID/FVD against a real panoramic video distribution, or a user study on in-the-wild inputs. For a generation paper, the absence of any perceptual evaluation is a notable gap.

- The 2D vs 3D paradigm framing is oversold. Generation still happens in ERP pixel space via CogVideoX's latent diffusion. What's actually 3D is the control signal (PPE) and the attention mask (sphere-aware). A more honest framing: 2D generation with 3D-aware conditioning and attention. The paper also claims that this paradigm addresses temporal forgetting, but spherical representation helps with spatial consistency per frame rather than temporal drift over long sequences. These are different problems.

Smaller Points:

- The baselines comparison is necessarily awkward given the task novelty. GenEX lacks a controllable checkpoint, and CameraCtrl/AC3D aren't trained on panoramic data. The authors make reasonable workarounds (perspective cropping, quality-only comparison), and it's hard to fault them for working on a problem where no directly comparable method exists.

- The resolution (480x960, resized from 480x720) is low for panoramic content. The authors explain that this preserves CogVideoX's pretrained priors, which is pragmatic, but it limits practical applicability and should be discussed as a limitation.

- The per-direction PSNR breakdown (Table G) is a nice analysis. It shows the method doesn't just improve in the front view but across all directions, validating the 360-degree consistency claim.

- The paper doesn't discuss how temporal forgetting (drift over long sequences) interacts with the spherical representation. Panoramic format gives you full spatial coverage per frame, but that's orthogonal to maintaining consistency over many frames. Worth at least a discussion.

- How does sphere-aware attention relate to correspondence-aware attention in other work? The paper could better position this connection. Both modulate attention based on geometric relationships, but sphere-aware attention operates on spatial position (where tokens are on the sphere) rather than ray-to-ray correspondence.

---

> ### Author Rebuttal · Authors · 2026-03-31
>
> **Visualization: https://panoworld-x.github.io/Video/**
>
> **Q1: User Study and Real-World Validation**
>
> We conducted a user study following 360DVD's protocol. 30 participants were shown panoramic videos (mixed test-set and in-the-wild inputs) and asked to rank all methods across four dimensions. The table below reports the percentage of times each method was ranked first:
>
> | Model | Graphics Quality ↑ | Frame Consistency ↑ | End Continuity ↑ | Content Distribution ↑ |
> |---|---|---|---|---|
> | 360DVD | 10.5% | 11.2% | 10.1% | 8.8% |
> | Imagine360 | 11.3% | 12.6% | 10.9% | 9.4% |
> | GenEX | 26.4% | 27.2% | 29.5% | 23.2% |
> | **Ours** | **51.8%** | **47.0%** | **56.5%** | **62.6%** |
>
> For real-world evaluation, we build a 100-image set from WEB360, Matterport3D, and self-collected panoramas.
>
> | Model | FID ↓ | FVD ↓ | Q-Align ↑ |
> |---|---:|---:|---:|
> | 360DVD | 92.31 | 167.39 | 0.487 |
> | Imagine360 | 87.24 | 135.03 | 0.492 |
> | GenEX | 82.31 | 142.35 | 0.413 |
> | **Ours** | **64.82** | **103.45** | **0.551** |
>
> Our method performs best on all metrics, showing strong real-world generalization (**full results on our project page**). We attribute this to our lightweight sphere-aware conditioning modules, which preserve CogVideoX's pre-trained visual priors.
>
>
> **Q2: 2D vs 3D Paradigm Framing**
>
> We agree and will adopt a clearer framing: prior methods are **2D-aware** (operating in ERP pixel space without explicit spherical modeling), while ours is **3D-aware** (injecting spherical geometry through PPE and Sphere-Aware Attention). We will revise the manuscript accordingly.
>
> Spatial consistency vs. temporal forgetting. The reviewer is right to flag this distinction. Our intended claim is the same: per-frame spatial errors accumulate over time and manifest as temporal inconsistency—these two issues are inherently linked. We will revise the wording to make this clearer.
>
> **Q3: Video Resolution**
>
>  To address this limitation in practice, we have applied video super-resolution models as a post-processing step to upscale the generated panoramic videos. The super-resolved results demonstrate significantly improved visual quality and are showcased on our anonymous project page. Importantly, this enhancement is orthogonal to our core contributions (PPE, Sphere-Aware Attention) and can be seamlessly applied to any panoramic video generation pipeline, making it a general and practical solution for resolution scaling.
>
>
> **Q4: Temporal Consistency and Long Sequences**
>
>  As described in Lines 257–271 (Eq. 4–5), Sphere-Attn operates in both spatial and temporal dimensions. The temporal component models camera movement as sphere rotation, transforming cross-frame points into a common reference frame before computing great-circle distance. This ensures the temporal attention mask tracks geometrically corresponding regions across frames — not merely co-located pixels — directly combating temporal drift. Ablation in Table F validates this: removing the dynamic temporal mask degrades PSNR from 19.34 to 18.26.
>
> To further validate long-sequence performance, we adopt an autoregressive extrapolation strategy where consecutive clips share overlapping frames at boundaries. Tail frames of the previous clip initialize the noise latents of the next clip, anchoring appearance and motion continuity. This produces continuous panoramic sequences of 15–20 seconds with minimal error accumulation. Results are on our project page.
>
> **Q5: Connection to Correspondence-Aware Attention**
>
> Both inject geometric priors into attention, but differ in two key aspects. **(1)** Ray-to-ray attention (e.g., MVDiffusion, DiffPano[NeurIPS24]) focuses on establishing multi-view correspondences via epipolar constraints, but does not address the inherent distortions of ERP projection—namely polar stretching and left–right boundary discontinuity. Our sphere-aware attention directly models spatial relationships on the sphere, simultaneously resolving both multi-view consistency and ERP-specific distortions. **(2)** Prior approaches only operate in the spatial domain (intra-frame). Our method jointly models spatial and temporal dimensions, ensuring cross-frame coherence alongside per-frame correctness. We will add this discussion.
>
> **Q6: Generation Quality at the Poles**
>
> We provide a latitude-wise PSNR analysis:
>
> | Latitude Band | Equator (±30°) | Mid (30°–60°) | Polar (60°–90°) |
> |---|:-:|:-:|:-:|
> | w/o PPE | 17.89 | 16.52 | 14.49 |
> | Full | 19.97 | 19.18 | 18.87 |
> | Δ Improvement | +2.08 | +2.66 | **+4.38** |
>
> PPE brings consistent improvements across all bands, with the largest gain at the poles (+4.38 PSNR), confirming that the spherical manifold mapping effectively resolves redundant geometric instructions at polar regions. Visual comparisons on our project page show coherent, artifact-free results.

---

> > ### Author Rebuttal · Reviewer_dweu · 2026-04-04
> >
> > Thank you for a thorough rebuttal that directly addresses all three questions with real numbers.
> >
> > The user study with 30 participants on mixed test sets and in-the-wild inputs is the most important addition. A 51.8% first-place preference rate on graphics quality, combined with the real-world FID/FVD results on 100 images from WEB360 and Matterport3D, substantially closes the evaluation gap I flagged. This is exactly what was needed.
> >
> > The latitude-stratified PSNR analysis is the most satisfying result in the rebuttal. The largest gain occurs at the poles (+4.38 dB vs +2.08 dB at the equator), directly validating the core PPE claim that spherical manifold mapping resolves redundant geometric instructions precisely where standard Plucker embeddings break down. This is a transferable insight that strengthens the paper's scientific contribution.
> >
> > The dynamic temporal mask ablation and autoregressive extrapolation strategy for long sequences partially addresses the temporal drift concern, though I would encourage the authors to include a more explicit discussion of this approach's limits in the final paper.
> >
> > I am raising my score to accept. Please ensure the user study results, real-world evaluation numbers, latitude-stratified analysis, and the revised framing of 2D-aware vs 3D-aware generation all appear in the camera ready version.

---

> > > ### Author Response · Authors · 2026-04-08
> > >
> > > Thank you for the positive feedback and for `raising the score`. We will carefully revise the final version to incorporate the key results and discussions based on your constructive suggestions.

---

### Official Review · Reviewer_18RU · 2026-03-12

**Soundness:** 3
**Presentation:** 2
**Significance:** 2
**Originality:** 2
**Overall Recommendation:** 4
**Confidence:** 4

**Summary:**

This work proposes a novel method for generating panoramic scene videos. The Exploration Panoramic Plucker Embedding is conceptually sound and effectively enhances trajectory-controllable panoramic video generation. Experimental results are impressive, demonstrating clear improvements over existing trajectory-controllable video generation and panoramic video generation methods.

**Compliance With Llm Reviewing Policy:**

Affirmed.

**Final Justification:**

My concerns have been addressed.

**Key Questions For Authors:**

I am also curious about the generalization ability of the proposed method. For example, how does it perform when given real panoramic images as input? Please provide more comparisons on real-world scenes, along with visualizations and analyses of more diverse scene types.

While trajectory-controllable video generation is a key claim, the paper lacks quantitative metrics and experiments evaluating scene consistency under varying camera trajectory motions.

**Limitations:**

yes

**Strengths And Weaknesses:**

Strengths:

Generating explorable and trajectory-controllable panoramic scene videos is an intriguing idea that partially addresses the limitations of conventional panoramic image generation, such as fixed viewpoints and restricted scene extensibility. This work further introduces a novel panorama dataset preparation pipeline in Unreal Engine, resulting in a collection of over 11K panoramic videos paired with corresponding trajectories.

Weaknesses:

1. The experimental design lacks a degree of fairness, making it challenging to reliably validate the effectiveness of the proposed method. In particular, the authors train their model on a self-constructed dataset, whereas comparative methods are not fine-tuned with corresponding training data. As a result, it is difficult to determine whether the observed advantages stem from the proposed method itself or from the use of larger or higher-quality training data. That said, the introduction of the dataset itself is a valuable contribution, so the authors are encouraged to conduct more equitable experimental comparisons, rather than being required to do so.

2. Could the authors provide more comparisons regarding training time, inference time, and model parameters? A comprehensive analysis of model training and rendering efficiency is needed, including inference duration for pre-trained models and other relevant time costs, as well as an analysis of model size.

---

> ### Author Rebuttal · Authors · 2026-03-31
>
> **Visualization: https://panoworld-x.github.io/Video/**
>
> **Q1: Fairness of Experimental Comparisons**
>
> We appreciate the reviewer's thoughtful concern. To enable a fair comparison, we fine-tune 360DVD and GenEX on our dataset (Imagine360 has not released its training code). Results are shown below:
>
> | Model | PSNR ↑ | SSIM ↑ | LPIPS ↓ | FID ↓ | FVD ↓ |
> |---|---|---|---|---|---|
> | 360DVD | 10.66 | 0.35 | 0.73 | 111.85 | 2049.21 |
> | 360DVD (finetuned) | 13.31 | 0.48 | 0.68 | 79.32 | 1239.13 |
> | GenEX | 16.12 | 0.59 | 0.42 | 42.22 | 1113.72 |
> | GenEX (finetuned) | 17.29 | 0.61 | 0.36 | 36.99 | 668.23 |
> | **Ours** | **19.34** | **0.63** | **0.24** | **28.01** | **467.18** |
>
> Both baselines improve after fine-tuning on our data. However, their overall performance still does not surpass ours, confirming that the advantages stem from our method design (PPE, Sphere-Aware Attention, Metric-Anchor Normalization) rather than data alone. This is further supported by the ablation study in Table 2, where each component contributes meaningfully on the same dataset.
>
>
> **Q2: Training and Inference Details**
>
> Our method introduces moderate overhead in memory and computation. The original CogVideoX-5B-I2V uses 5 GB memory and takes about 200 s to generate a 49-frame video, whereas our method uses 8 GB and takes about 280 s on a single H20 GPU. The training cost increases from 63 GB to 78 GB, and takes about 3--4 days. We consider this trade-off worthwhile because it substantially improves spherical consistency, especially left-right boundary continuity, which is critical for immersive viewing and helps avoid visible vertical seams.
>
> Compared with GenEX (2.25B), which takes about 100 s to generate 15 frames, our method remains competitive after frame normalization: 6.67 s/frame for GenEX versus 5.71 s/frame for ours, while achieving clearly better quantitative results and visual quality. Since the other baselines perform substantially worse than GenEX in generation quality, efficiency comparisons with them would be less informative. We therefore focus on GenEX as the strongest and most meaningful baseline for efficiency comparison.
>
>
> **Q3: Generalization to Real Panoramic Images**
>
> To evaluate our model's generalization to real-world data, we construct a real-world evaluation set comprising 100 panoramic images from the WEB360 dataset (a public real-world panoramic collection) with additional real-world panoramic images collected by ourselves to maximize the diversity of scene categories. As shown in the table below, our model achieves the best FID, FVD, and Q-Align scores among all compared methods.
>
> | Model | FID ↓ | FVD ↓ | Q-Align ↑ |
> |---|---|---|---|
> | 360DVD | 92.31 | 167.39 | 0.487 |
> | Imagine360 | 87.24 | 135.03 | 0.492 |
> | GenEX | 82.31 | 142.35 | 0.413 |
> | **Ours** | **64.82** | **103.45** | **0.551** |
>
> We attribute this strong real-world generalization to our design philosophy: the proposed sphere-aware components (PPE and Sphere-Aware Attention) are injected as lightweight conditioning modules that adapt the model to the panoramic modality without disrupting the rich visual priors learned by the CogVideoX backbone during large-scale pre-training. We have also added diverse scene visualizations on our anonymous project page, covering outdoor landscapes, indoor rooms, urban streets, natural terrains, and cultural landmarks, to demonstrate broad generalization across scene types.
>
>
>
> **Q4: Scene Consistency Under Varying Trajectories**
>
> Tables 1 & 2 already report **R_err** and **T_err**, measuring the deviation between estimated and ground-truth camera poses. Our method achieves the lowest errors among all compared methods. To further evaluate robustness, we break down results by trajectory type:
>
> | Trajectory Type | R_err ↓ | T_err ↓ |
> |---|---|---|
> | Straight Forward | 0.053 | 0.062 |
> | Turn Left | 0.064 | 0.076 |
> | Turn Right | 0.063 | 0.075 |
> | Multi-Turn | 0.071 | 0.082 |
> | Variable Speed | 0.062 | 0.074 |
> | **Overall** | **0.061** | **0.073** |
>
> R_err and T_err remain **consistently low** across all five types, confirming robustness to different camera directions and speeds. This stems from PPE's explicit spherical geometric guidance and Sphere-Aware Attention's spatiotemporal coherence. Qualitative evidence in Fig. D further shows consistent results across diverse exploration routes.

---

> > ### Author Rebuttal · Reviewer_18RU · 2026-04-03
> >
> > My concerns have been addressed. Will the authors release the code and model weights?

---

> > > ### Author Response · Authors · 2026-04-03
> > >
> > > Thank you for your positive feedback—**we’re glad the concerns have been addressed.**
> > >
> > > We plan to open-source both the code and model weights upon acceptance, and are undergoing internal legal and compliance review of the data and code as part of the standard release process.
> > >
> > > We appreciate your patience and support,  **and would be grateful if you could consider updating your score to reflect the addressed concerns.**

---

### Decision · Program_Chairs · 2026-04-30

**Decision:**

Accept (spotlight)

**Comment:**

This paper received strong positive ratings (2 Accepts, 2 Weak Accepts). The reviewers acknowledged that the paper introduced an innovative 3D-aware paradigm to panoramic video generation. Initial concerns centered on the lack of evaluation on real-world or dynamic scenes. The authors' rebuttal successfully addressed these critiques. Consequently, all reviewers expressed high satisfaction with the rebuttal, resulting in strong positive scores across the board. Given this, the AC recommends acceptance.